# Understanding Estimations of Magnitudes: An fMRI Investigation

**DOI:** 10.3390/brainsci12010104

**Published:** 2022-01-12

**Authors:** Sarit Ashkenazi, Yarden Gliksman, Avishai Henik

**Affiliations:** 1The Seymour Fox School of Education, The Hebrew University of Jerusalem, Jerusalem 91905, Israel; 2Department of Behavioral Sciences, Ruppin Academic Center, Emek Hefer 40250, Israel; yarden.gliksman@gmail.com; 3Department of Psychology and Zlotowski Center for Neuroscience, Ben-Gurion University of the Negev, Beer-Sheva 84105, Israel; henik@bgu.ac.il

**Keywords:** cognitive estimation, numerical estimation, executive functions, angular gyrus

## Abstract

The current study examined whether discrete numerical estimation is based on the same cognitive process as estimation of continuous magnitudes such as weight and time. While the verbal estimation of numerical quantities has a contingent unit of measurement (e.g., how many cookies fit in a cookie jar? _X_ cookies), estimation of time and weight does not (e.g., how much time does it take to fill a bath with water? _X_ minutes/hours/seconds). Therefore, estimation of the latter categories has another level of difficulty, requiring extensive involvement of cognitive control. During a functional magnetic resonance imaging (fMRI) scan, 18 students performed estimations with three estimation categories: number, time, and weight. Estimations elicited activity in multiple brain regions, mainly: (1) visual regions including bilateral lingual gyrus), (2) parietal regions including the left angular gyrus and right supramarginal gyrus, and (3) the frontal regions (cingulate gyrus and the inferior frontal cortex). Continuous magnitude estimations (mostly time) produced different frontal activity than discrete numerical estimations did, demonstrating different profiles of brain activations between discrete numerical estimations and estimations of continuous magnitudes. The activity level in the right middle and inferior frontal gyrus correlated with the tendency to give extreme responses, signifying the importance of the right prefrontal lobe in estimations.

## 1. Introduction

How do we estimate magnitudes? Do we use different mechanisms and different neural regions to estimate different domains such as weight, time, or numerical quantities? The current study suggests a new cognitive estimation task (CET) to examine this issue.

One of the accepted views in the field of numerical cognition suggested an innate mechanism in charge of estimation, that is, the approximate number system (ANS) [1,2,3]. Converging evidence from infants, preschool children and adults, as well as non-human primates, suggests that the representation of approximate discrete quantities is a foundational ability that is supported by the intraparietal sulcus (IPS) in the posterior parietal cortex [4,5,6,7]. This system is devoted to approximating how many non-symbolic stimuli are presented and comparing between arrays of non-symbolic stimuli. The ANS view suggests the cognitive system is sensitive to discrete entities of stimuli (number sense) independently from non-numerical continuous entities (i.e., visual properties (e.g., density) and non-visual properties (e.g., duration or weight of stimuli)).

### 1.1. From Continuous Properties of Physical Stimuli to Numerical Processing

However, discrete and continuous properties of physical stimuli are most of the time-correlated [8,9]. For example, choosing a basket with more fruits can be based not only on the number of the fruits (discrete property), but also on their weight (continuous properties). Alternatives to the ANS view, recent theories suggested that continuous properties have an important role [8,9,10,11]. Interestingly, a theory of magnitude (ATOM; Walsh, 2003) proposed a common innate mechanism for the processing of discrete and continuous quantities [12] of time, space, and numerical quantity [13].

Additionally, some very early developmental indications suggest that the understanding of continuous variables develops before the understanding of numerical discrete quantities and affects it. For example, in his classical experiment, Piaget [14] presented children with two rows that contained the same number of coins. In the first phase, the two rows had the same surface area and the experimenter asked the child whether the rows contained the same or a different number of items. The child usually answered “the same”. In the second phase, the experimenter changed the spacing between items in one of the rows in front of the child and then asked the same question again. Young children were likely to answer that the row with the larger spacing had more items than the other row.

Accordingly, Leibovich et al. [9] suggested a developmental model that explains the relation between the processing of continuous entities and discrete entities of a given stimulus and emphasizes the importance of language in creating symbolic processing. This model postulates that numerical sensitivity develops from repetitive comparisons of continuous magnitudes of groups of objects. In most of the incidents, there are positive and high correlations between continuous properties of the stimuli and numerical values, and these correlations with language (i.e., number words) help a child acquire discrete numerical understanding. Counting objects or naming quantities teach children that the same number word can represent objects of various sizes, areas, and other non-numerical entities, and direct a child’s attention to numerical discrete values and counting. In the last stage, children develop the ability to deal with the case where numerical values and non-numerical entities are not correlated or are negatively correlated. Domain-general abilities, such as cognitive control, help children deal with these cases of conflict between numerical and non-numerical entities of a given stimulus. Hence, if children first understand non-numerical entities and only then, with the acquisition of words, children understand numerical entities, the neural representation of non-numerical and numerical entities should be dissociable [5,6,7].

### 1.2. Cognitive Estimation, the Role of Executive Function and Quantity Estimation

Cognitive estimation is a process that is critical to many daily life activities, such as estimating the weight of a suitcase, estimating the time of driving to arrive at work, or the cost of several products in a store. One applied task that was designed to examine estimation is CET. CET examines verbal estimations of continuous magnitudes and numerical entities. It provides estimations for answers to questions for which relevant exact knowledge, but not the specific answer, is available to the subject [15] (e.g., how many cookies fit in a cookie jar?). CET was originally designed to evaluate patients with executive function (EF) weakness, who usually give very extreme estimations in CET [15]. Indeed, some estimations do not have an exact answer or a learned solution strategy; they require multiple unknown stages to solve. Multiple unknown stages for a solution require the involvement of numerous aspects of EF, such as working memory, attention, cognitive control, and planning [15]. The dorsolateral prefrontal cortex (DLPFC) is implicated in EF [16,17], and specifically was found to be of importance for CET [18].

Importantly, CET also requires quantity understanding. Recently, Bisbing et al. [19] compared CET results between patients with weakness in EFs due to frontal damage, to patients with weakness in numerical processing due to right inferior parietal damage. The authors found that deficits in CET could be either a result of frontal damage in the right lateral prefrontal and orbital frontal cortices (brain regions that are related to EFs) or a result of damage in the right inferior parietal cortex (brain regions that are related to numerical processing). Even though these results are informative, the role of EF and numerical estimation in CET and the underlying neural mechanism of CET in the normative population are still largely unknown.

EF is an umbrella term for a set of high-order cognitive processes that play a critical role in learning (including; planning, problem-solving, cognitive flexibility, working memory, mental control, inhibition, and self-monitoring). Working memory is one of the most investigated EF tasks and includes: (1) the phonological loop for maintenance of verbal information—the ventrolateral PFC (VLPFC) is thought to be involved in short-term maintenance of verbal information [20]; (2) the central executive that is involved in the manipulation of information—the dorsolateral PFC (DLPFC) has been most commonly implicated in maintaining information and has also been suggested to be involved in planning [20]. The anterior cingulate cortex (ACC), which is part of the medial PFC, was found to be involved in conflict monitoring [20]. For example, the ACC is involved when there are two or more competing behaviors that may be triggered by events in our environment, and one of the behaviors should be inhibited.

### 1.3. CET of Continuous Magnitude or Discrete Numerical Information: The Usage of Measurement Units

The use of measurement units is hard because units of measurement impose discrete units onto continuous measurements [21]. For example, in weight and time estimation, which are continuous variables, when the units of measurement are not part of the question (how much time does it take to fill a bath with water? _X_ minutes), understanding the unit of measurement, fitting it to the estimation, together with the mixture of continuous entities and discrete measurement units, all add extra steps and difficulty and require domain-general processes such as EF. However, in discrete numerical quantities estimation, such as the number of cookies or children, the unit of measurement is part of the question and is discrete (i.e., “cookies”) and thus very intuitive to use. For example, how many cookies fit in a cookie jar? _X_ cookies. In the estimation of continuous variables, working memory and cognitive control are needed to remember and to manage the multiple stages leading to a solution. Namely, even if the estimations of all magnitudes originate from a common mechanism [13], there is a need to use different units of measurements for the various contexts of magnitude. Silverman and Ashkenazi [22] examined the developmental effect of usage of units of measurement in CET, comparing groups of 10- and 12-year-old children to adults. Different developmental trajectories for different estimation categories were found. Children’s estimations were more extreme, relative to adults, in weight and time evaluations, but comparable to adults in quantity evaluations. The authors concluded that CET questions that require estimations of continuous entities using discrete measurement units are more difficult for children due to higher involvement of executive functions, and children have less experience applying them in daily life [22]. In another study, Ashkenazi and Tsyganova [23] found that different estimation categories had different relations to domain-general abilities (e.g., IQ, working memory) and numerical abilities. Specifically, the results of structural equation modeling found shared variance between weight, time, and distance estimations, estimations of continuous entities using discrete measurement units, and no shared variance with numerical estimation discrete measurement, which does not require the usage of external units of measurements. The authors suggested that numerical estimation, which is discrete (i.e., “cookies”) and thus very intuitive to use, represents a more “pure” estimation. In line with this view, the authors found that numerical estimation was predicted by preverbal innate quantity understanding (approximate number sense) and working memory, whereas time estimations were supported by IQ [23].

### 1.4. The Current Study

The current study presents a new approach to test CET that is based on our previous studies [22,23] and customized to be used during fMRI. Our study aims to answer several questions. CET was originally designed to test EF weakness, however, the underlying neural mechanism of CET in the normal population and the involvement of frontal activation are largely unknown. Hence, the main goal of the present study was to understand the underlying neural mechanism of CET is. We hypothesized that the prefrontal, orbital frontal, and inferior parietal cortex would be activated during CET.

Moreover, we wanted to evaluate the respective role of EF and numerical estimation in the CET process. We examined brain regions that were activated during CET and explained individual differences in CET estimations. Prefrontal or orbital frontal activation might indicate involvement of EF in CET, while inferior and posterior parietal cortex activation might indicate involvement of numerical estimation in CET [19].

Another important question of the current study was whether the different types of magnitudes (i.e., weight, time, and numerical quantities) have shared or distinct cognitive representations. To examine this, we presented participants with estimation questions regarding those magnitudes. The use of different units of measurement varies among various contexts of magnitudes. While for numerical quantity estimation, the unit of measurement is part of the question and both are discrete (unit of measurement and magnitude), for time and weight estimation, which are continuous magnitudes, the use of independent units of measurement is required, such as minutes, hours or kilograms (discrete unit of measurement). Hence, we hypothesized that numerical quantity estimations would be distinct from weight estimation and time estimation, which would recruit more regions related to cognitive control.

## 2. Materials and Methods

### 2.1. Participants

Twenty-two students from Ben-Gurion University of the Negev, Beer-Sheva, Israel, were recruited to participate in the experiment. All of the participants were university students and most of them were studying psychology. All participants were right-handed and monolingual native Hebrew speakers, with intact or corrected vision and no reported learning disabilities or attention deficits. Participants were compensated for their participation in the experiment with a monetary reimbursement ($25). Four participants were excluded from the analysis due to either (a) excessive motion during scanning (more than 2-mm deviations from the first image collected and/or more than a 1-mm deviation between one functional image to the next functional image—three participants), or (b) low accuracy rates in one or more conditions (i.e., less than 50% accuracy).

For the remaining 18 (12 women) participants, the average age was 22 years and 9 months (*SD* = 6 years and 8 months). All experimental procedures were approved by the Helsinki Committee of Soroka Medical Center, Beer Sheva, Israel.

### 2.2. Stimuli and Procedure

#### 2.2.1. Behavioral Pilot: Creating CET Questions for the fMRI

CET is a task where participants are asked to give an estimation of a question with a range of plausible solutions. Three estimation categories and one control condition were examined. Estimation was examined in weight, time, and numerical estimation categories. The control condition included questions with exact knowledge answers, and served as the baseline condition in the current study (e.g., How many days are there in a week?).

The pilot study was conducted to create a range and percentiles for the estimation questions. The questionnaire included background questions (e.g., age, gender, years of education, drugs and medicine usage). Participants were then presented with questions in which they were instructed to estimate the correct answer. The 64 questions were composed of four types of questions. Each question was presented with the units of measure of the expected answer in brackets (e.g., exact knowledge: How many letters are there in the Hebrew alphabet? (letters); weight: What is the weight of the fattest man in Israel? (kilograms); time: How much time does it take for flowers in a vase to dry? (days); numerical estimation: How many cookies can be contained in a cookie jar? (cookies)). Participants were instructed to type their numerical answers using the keyboard. We created the questionnaire using google-docs. A link to the questionnaire was sent to all first-year psychology students in the Hebrew University of Jerusalem, with a letter that thanked them in advance for taking part in the research. Participation was voluntary, with no identification of the participant. The results were based on the answers of 48 students. We computed four quartiles of the estimations, which were the basis of the ranges in the fMRI study (see below).

#### 2.2.2. fMRI CET Task

The tasks were programmed in E-Prime 2.1. The fMRI experiment was an event-related design with four task conditions—exact knowledge, weight, time, and numerical estimation—of 16 trials each, divided into two blocks (32 trials per block). As in the pilot study, in each trial, participants were presented with a question and unit of measure of the expected answer in brackets. Participants were instructed to estimate the answer to the presented question and were told in advance that for most questions, any estimated answer would be correct, meaning that there was not just one, unitary correct estimation but a range of estimations were correct. Then participants were presented with four ranges of answers (according to the quartiles calculated in the pilot study) and were asked to choose the range that included their estimation (e.g., for a weight question—1. Below 112 kg; 2. Between 112–360 kg; 3. Between 361–520 kg; 4. More than 520 kg). The questions and answers were presented in Courier New font, size 72. Each trial began with a fixation point in the middle of the screen, presented for 1 s, followed by a black screen for 300 ms. Then, an estimation question was presented for 8 s, followed by the four ranges presented for 3 s. During the presentation of the ranges, participants responded using a response box. Only the first eight seconds were included in the analysis of the fMRI time window, to avoid differentiating response time (RT) between conditions and motor activity. RT was measured in milliseconds from the onset of the ranges until the participant’s keypress. The next trial started after a jitter period that varied between 4997 ms and 20,246 ms, with an average of 13,300 ms. The total length of the experimental block was 13 min and 35 s (see Figure 1 below). The independent variable was category (exact knowledge, weight, time, and numerical estimation). Only correct trials were analyzed (see Appendix A for the percentage of correct trials in each of the categories).

#### 2.2.3. fMRI Acquisition

Whole-brain functional data were acquired using a 3-Tesla Philips ingenia MRI scanner using a gradient echo-planar imaging (EPI) sequence (TR = 2 s; TE = 35 ms; flip angle = 90 degrees). The order of imaging acquisition was ascending—interleaved, covering the entire brain of participants. For each functional volume, 33 slices (3 mm thickness, Field of view (FOV) = 230 mm × 245 mm × 115 mm, matrix = 76 × 83) were collected, resulting in a spatial resolution of 3 mm isotropic voxels. Each scanning session included the acquisition of a T1-weighted three-dimensional volume (voxel dimension = 1 mm × 1 mm × 1 mm) for co-registration and anatomical location of functional data.

#### 2.2.4. fMRI Analysis

Data analysis was conducted using the BrainVoyager QX software package (Brain Innovation, Maastricht, The Netherlands). The first four images of each functional scan were discarded. Preprocessing of functional scans included 3D motion correction, slice scan time correction, and removal of low frequencies, up to three cycles per scan (linear trend removal and high-pass filtering). The anatomical and functional images were transformed to the Talairach coordinate system using trilinear interpolation. For whole-brain group analyses, preprocessing of functional scans additionally included an 8 mm spatial smoothing. The whole-brain group contrasts were computed using a random effect generalized linear model (GLM) analysis, corrected for multiple comparisons with a false discovery rate of q(FDR) < 0.05.

#### 2.2.5. fMRI Multi-Voxel Pattern Analysis (MVPA)

To further investigate cognitive estimation and different categories, we conducted a multi-voxel pattern analysis. We calculated a whole-brain support vector machine (SVM)-based searchlight mapping [24] between the activations for the different estimation categories. We reported accuracy rates greater than 70%, and cluster levels greater than 400 voxels.

## 3. Results

### 3.1. Behavioral Results

First, we describe the response frequencies according to category and then the analysis of variance (ANOVA) with category as a within-subject variable—performed once with the number of extreme responses as the dependent variable and once with RT. Appendix A represents a descriptive statistic of the fMRI task, behavioral results.

#### 3.1.1. Responses Frequencies

Figure 2 presents the percentiles of responses by category and answer (i.e., percentiles of ≤24, 25–49, 50–74, and ≥ 75). As can be seen, most of the answers were in the second or third percentile range for the numerical estimation category (95%) and the weight category (93%); the time category (90%), and 82% of the responses in the exact knowledge condition fell in the third percentile range.

#### 3.1.2. Extreme Responses

We calculated the number of extreme responses (more than percentile 75 and less than percentile 25) by category using an ANOVA with category as a within-subject variable.

There was a main effect of category, *F*(3, 48) = 2.28, *p* < 0.05. The number of extreme responses was lower in the exact knowledge condition (0.24, *SD* = 0.56) compared to the time condition (1.12, *SD* = 0.99), *t*(17) = −3.45, *p* < 0.001. Additionally, the number of extreme responses was lower in the numerical estimation condition (0.53, *SD* = 0.87) than in the time condition, *t*(17) = −2.28, *p* < 0.05. There were no significant differences between number of extreme responses in the weight category (0.65, *SD* = 1.23) and any other category (min *p* = 0.12). All the other comparisons were not significant (Figure 3).

#### 3.1.3. RT

There was a main effect of category, *F*(1, 14) = 13.6, *p* < 0.002. RTs were lower in the exact knowledge category (1589 ms, *SD* = 217) compared to the numerical estimation (1806 ms, *SD* = 149), time (1805 ms, *SD* = 152) and weight (1824 ms, *SD* = 167) categories, *t*(16) = −4.89, *p* < 0.001; *t*(16) = −4.5, *p* < 0.001; *t*(16) = −4.2, *p* < 0.001, respectively. There were no significant differences between the estimation categories (i.e., between numerical estimation, time, and weight).

### 3.2. fMRI Results

#### 3.2.1. Whole-Brain Univoxel Analysis

##### Brain Activations Related to CET

To investigate the impact of CET on brain activation, we performed a whole-brain *t*-test statistic pitting the brain signal related to CET against the brain signal associated with the exact knowledge category across all estimation types (numerical estimation, time, and weight). The results of this analysis (see Table 1 for all contrasts) revealed nine regions that were more activated in the estimation than the exact knowledge category—the right anterior and posterior cingulate, right caudate, right precentral gyrus, left insula, left and right lingual gyrus, and left and right cerebellum. Two brain regions were more deactivated during CET compared to activation associated with the exact knowledge category: the left angular gyrus and the right supramarginal gyrus (Figure 4 and Table 1).

##### Numerical Estimation vs. Weight

No significant differences were found.

##### Numerical Estimation vs. Time

Results of this analysis revealed greater activation in the numerical estimation versus time categories in the left and right lingual gyrus. Additionally, greater deactivations were found in the numerical estimation versus time categories in the right inferior frontal gyrus (Figure 5 and Table 1).

#### 3.2.2. Whole-Brain Multi-Voxel Pattern Analysis (MVPA)

Brain Activations Associated with Numerical Estimation vs. Weight and Time Estimation (Separately)

We calculated a whole-brain SVM-based searchlight mapping between the brain activation of the numerical estimation versus the brain activation of the weight and time categories.

##### Numerical Estimation vs. Time

Results revealed differences in various frontal regions: the right inferior frontal gyrus, right and left middle frontal gyrus (R Brodmann Area (BA) 6, 8 and L BA 8, 9), right and left superior frontal gyrus (R BA 6, L BA 8), right and left anterior cingulate and left precentral gyrus. Moreover, this analysis revealed differences in the left angular gyrus and left lingual gyrus and thalamus (Figure 6A and Table 2).

##### Numerical Estimation vs. Weight

Results revealed differences in the right middle frontal gyrus, right postcentral gyrus, left putamen, left caudate, and the right cerebellum (Figure 6B and Table 2).

Correlation Analysis on the Contrast of CET and Exact Knowledge with Extreme Responses

Weakness in the CET paradigm is usually associated with extreme responses [15]. Hence, here we examined the correlation between the individual tendency to give extreme responses and brain activation. We calculated a whole-brain covariate model (ANCOVA) with the number of extreme CET responses (more than percentile 75 and less than percentile 25) as the independent variable and brain activity level during estimations as to the dependent variable. The results indicated positive correlations between the activity of the right inferior frontal gyrus (BA 46), *r*(16) = 0.74, *p* < 0.01, the right middle frontal gyrus (BA 9), *r*(16) = 0.78, *p* < 0.01, and the left superior temporal gyrus (BA 22), *r*(16) = 0.81, *p* < 0.01, (Figure 7 and Table 3) and number of extreme responses.

## 4. Discussion

The current study examined a fundamental question in the numerical cognition field—whether discrete numerical estimation is based on the same cognitive processes as estimation of continuous magnitudes such as weight and time. Current theoretical models emphasize the developmental role of language in the understanding of numerical discrete quantity versus continuous magnitude [21]. Hence, here we targeted verbally mediated estimations using CET.

We designed a new method to test CET during fMRI scanning. First, we created a CET with three categories: weight, numerical estimation, and time. Then, we carried out this task on a pilot sample without fMRI. Based on the results of the pilot sample, we created a CET multiple-choice questionnaire with ranges of answers provided, to avoid verbal responses when carried out during an fMRI scan. Additionally, to avoid a response bias during the fMRI scan, we first presented the CET question and then the possible solutions (see Figure 1).

One of the dominant roles of language in estimation is based on the usage of measurement units that impose discrete units onto continuous measurements [21]. Hence, even if the estimations of all magnitudes originate from a common mechanism [13], there is a need to use different units of measurements for the various contexts of magnitude. Whereas continuous magnitudes require the usage of measurement units, discrete numerical quantities do not require the usage of external measurement units. Hence, verbal estimation of continuous magnitudes such as time and weight should manifest more cognitive control than estimations of numerical quantities, due to imposed discrete units onto continuous magnitudes [21].

Importantly, understanding of units of measurements of time is more complex than understanding units of measurements of other continuous magnitudes, such as weight, because using units of measurements of time do not follow the base-ten structure of the symbolic numerical representation, but rather a base-sixty structure (e.g., hours).

In line with this view, multi-voxel pattern analysis found activation differences between numerical estimation and time estimations in various frontal regions including bilateral middle and inferior frontal gyri (BA 6, 8, and 9). Moreover, MVPA found activation differences between numerical estimations and weight estimations in the right medial frontal gyrus (BA 9).

CET was originally designed to evaluate patients with EF weakness, and an extensive study was done on CET in atypical populations; however, studies testing CET and the neural correlations of CET in the normal population are rare. Hence, another goal that guided the present study was to test the neural correlations of CET in the normal population. The results indicated that in the normative population, CET elicits activity in multiple regions. The loci of activity can be categorized into three major brain systems—the visual system (bilateral lingual gyrus), the parietal regions (left angular gyrus and right SMG gyrus), and frontal regions (cingulate gyrus and the inferior frontal cortex)—suggesting that CET involves brain regions that are associated to complex problem solutions, including numerical estimation, verbal understanding, and attention.

In atypical populations, CET is usually used to look for extreme responses, which are an indication for non-effective estimations. To test the role of extreme responses in the typically developed population, we correlated the number of extreme responses and brain activity during CET. Activity level in the right middle and inferior frontal gyrus correlated with the tendency to give extreme responses, demonstrating the importance of the right prefrontal lobe in CET. Moreover, the correlation between activity level in the right middle and inferior frontal gyrus, domain-specific brain regions that are related to EF, and the tendency to give extreme responses, demonstrate that EF plays a significant role in CET, especially, in the tendency to give extreme responses.

The last goal of the present study was to evaluate the respective role of numerical estimation and EF in the CET process. It is widely accepted that to solve CET questions, one is required to activate EF (including working memory, planning, inhibition, shifting, and attention) [15,25]. More current approaches, however, also emphasize the critical role of numerical estimation in the solution of CET [19]. CET elicits activity in multiple regions related to EF such as the right anterior cingulate (BA 6) and the left inferior frontal gyrus (BA 13). Importantly, CET elicits activity in multiple regions related to numerical processing, mainly in regions in the inferior parietal lobe, including the left angular gyrus and the right SMG, brain regions that are involved in verbally mediated numerical processing [26].

### 4.1. CET Elicits Activity in the Frontoparietal Network Related to EF and Numerical Cognition

In the parietal lobe, the bilateral IPS in the posterior parietal cortex is believed to hold innate, preverbal approximate quantity representation [7,26,27]. This representation is commonly viewed as being strongly involved in estimation processes [26,27]. However, contrary to the expected role of the IPS in estimation, in the parietal lobe, only the left angular gyrus and the right SMG, but not the IPS, were found to differentiate between estimations and exact knowledge. Within the domain of numerical cognition, two different representational codes have been proposed for the processing of numerical information. One code, supported by the activation of the angular gyrus and the SMG, is language-based and is used for exact calculations and well-learned verbal arithmetic operations. The other code, supported by the activation of the IPS, is a semantic visuospatial-based format, used for the representation of quantities on a mental number line, and required for the comparison of quantities and approximation processes [26].

The left angular gyrus, which is an essential part of the language-based code, was found to be more deactivated during estimation than exact knowledge [26,27]. Some studies have reported greater responses in the left angular gyrus during more automated calculation tasks [28,29]. A few studies have reported relative decreases or deactivation in the left angular gyrus during a simple calculation task [30,31]. Importantly, this deactivation was found to be related to individual differences in performance in the calculation task [31]. The deactivation elicited in the angular gyrus in the present study hints that CET requires exact verbal calculation rather than innate preverbal approximate quantity representation.

Another important area whose activation was modulated during CET was the right SMG. The right SMG was previously found to be related to CET: in a study that tested patients with parietal damage, the size of the lesion in the right inferior parietal cortex (approximately next to the SMG) was found to predict extreme performance in CET [19]. The right SMG elicited activity in multiple verbally mediated tasks including verbal working memory [32], phonological processing [33], and calculations [31]. However, the right SMG was involved in attentional tasks more than in multiple verbally mediated tasks [34], and its activation was found to be modulated by task difficulty (i.e., short-term visuospatial working memory load [35]). The temporal-parietal junction refers to the region of the cerebral cortex that lies along the boundary of the temporal and parietal lobes that border the right SMG and is involved in the stimulus-driven selection of important objects in the environment [36]. The right temporal-parietal junction is strongly activated by behaviorally important objects outside the current focus of attention that cause attention to be reoriented [37]. Deactivation of this region can serve as a filtering mechanism, suppressing irrelevant information when the task that is being carried out demands a lot of attention. For example, in a visual search task, deactivation of the right SMG was larger for successful trials (hits) than unsuccessful trials (misses) [38]. The right SMG was deactivated in the current study for estimations but not for exact knowledge; similarly, it was deactivated during calculations [30,31]. Hence, the deactivation found in the current study could be the result of domain-specific verbal calculation demands or domain-general task difficulties involved in estimations compared to exact knowledge [39].

Frontal lobe activation is traditionally related to EF demands. CET requires multi unknown stages to be solved that require the involvement of EF, such as working memory, attention, and planning. Hence, it comes as no surprise that CET elicits activation in the anterior and posterior cingulate gyrus bilaterally—an important region related to EF, error monitoring [40], integration of information [41] and resolving conflict [42]. Another brain region that was found to be involved in CET was the inferior frontal gyrus, which is associated with monitoring simple rules or few rules [34].

### 4.2. Distinct Brain Activation to Discrete Numerosity and Continuous Magnitude Estimations

Here we examined the effect of CET for different categories. While questions that estimate numerical quantities have no unit of measurement, CET of time and weight require the use of a unit of measurement. Therefore, estimation of the latter categories has another level of difficulty, requiring extensive involvement of cognitive control.

Both behavioral and multi-voxel pattern analysis confirm this assumption. Regarding the multi-voxel pattern analysis, extensive prefrontal cortex regions differentiate between numerical and time estimation, including the bilateral inferior and middle frontal gyrus and the bilateral anterior cingulate. These activated areas in the prefrontal cortex involve EF [43].

Note, however, that the widespread differences between the activation elicited by numerical and time estimation were not found for the comparison between weight and numerical estimation. The comparison between weight and numerical estimation using multi-voxel pattern analysis elicited a different set of regions. In the prefrontal lobe, only the right middle frontal gyrus differentiated between the conditions. We suggest that while both time and weight require the usage of units of measurements, the use of units of measurements of time is unique compared to all the other units of measurements. While units of measurements of weight, for example, follow the base-ten structure (1 kg equals 1000 g or 1 g equals 1000 milligrams), the time units of measurement do not present internal consistency (1 day equals 24 h and 1 h equals 60 min).

The present results demonstrate that CET is not a unified construct and that the estimation category plays a significant role in the activation observed.

### 4.3. Predicting Individual Differences in CET

Additionally, we wanted to find regions that could predict individual differences between participants in CET. We created a regression model that combined the behavioral data (number of extreme responses in the percentile range ≤ 24 or ≥ 75) and brain activity during CET.

As was found in a previous study with groups of patients [19], the present results revealed that in the normal population, the activity levels in the right middle frontal gyrus (BA 46) and right inferior frontal gyrus (BA 9) were significantly correlated to task performance, demonstrating the importance of the right prefrontal lobe in CET.

## 5. Conclusions

Distinct brain activation characterized different estimation processes, demonstrating that estimation of various discrete and continuous magnitudes is not a unified construct. The results demonstrate that various magnitudes are processed in separate brain pathways, based on the need to use units of measurements or not. Verbal estimations of continuous magnitude but not discrete numerical values involve the usage of units of measurement that require extensive cognitive control and recruitment of widespread frontal networks.

An additional goal of the present study was to examine the neural correlations of CET in the typical population. We found a few interesting results. First, the frontoparietal network, which is involved both in EF and in numerical estimation, is strongly activated during CET. Second, we found that brain regions related to the verbal representation of numerical information are involved in CET. Last, we found relations between the behavioral tendency to give extreme estimations and the activity level in the right middle and inferior frontal gyrus, demonstrating the importance of the right prefrontal lobe in CET.

## Figures and Tables

**Figure 1 brainsci-12-00104-f001:**
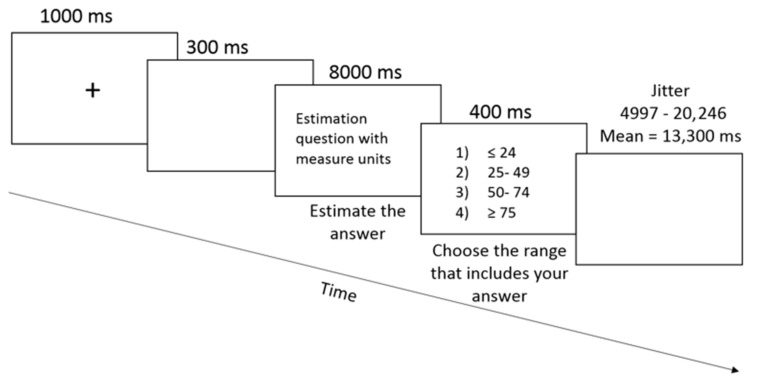
A trial in the cognitive estimation task.

**Figure 2 brainsci-12-00104-f002:**
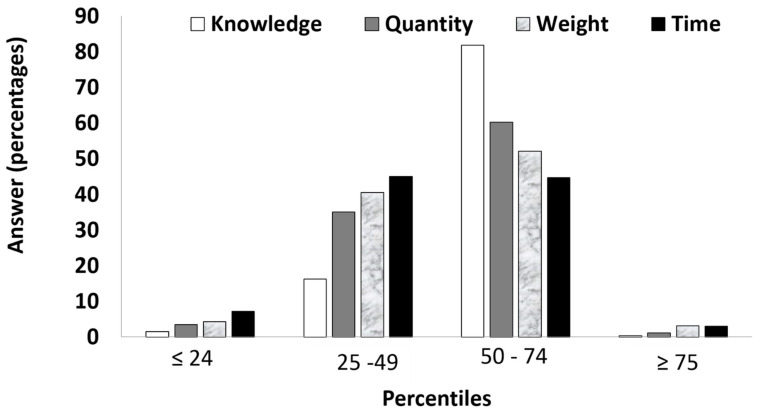
Percentiles of responses by category and answer.

**Figure 3 brainsci-12-00104-f003:**
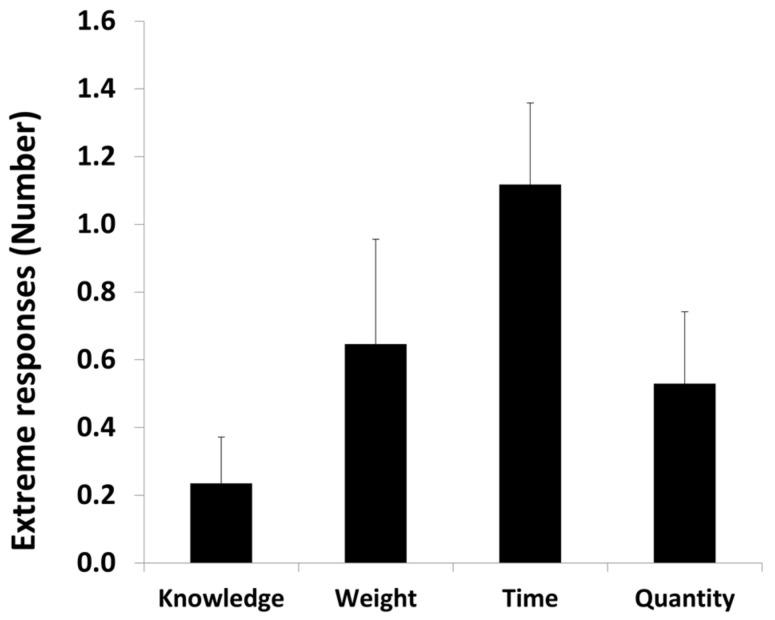
Number of extreme responses as a function of category.

**Figure 4 brainsci-12-00104-f004:**
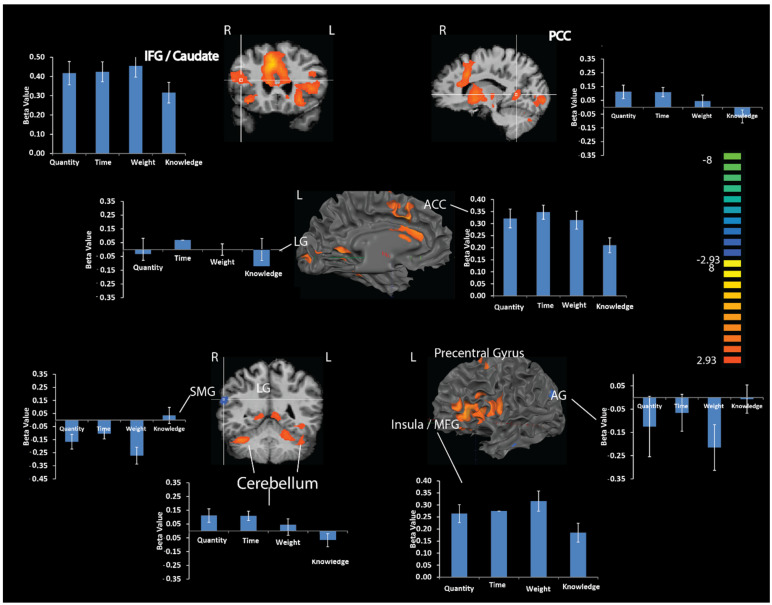
Brain activations associated with numerical estimation vs. weight or time estimation. To unravel the brain regions that showed significant brain activation differences between the estimation categories, we calculated a whole-brain *t*-test between the brain activation of the numerical estimation (discrete quantity) versus the brain activation of the weight and time category (continuous quantities), separately.

**Figure 5 brainsci-12-00104-f005:**
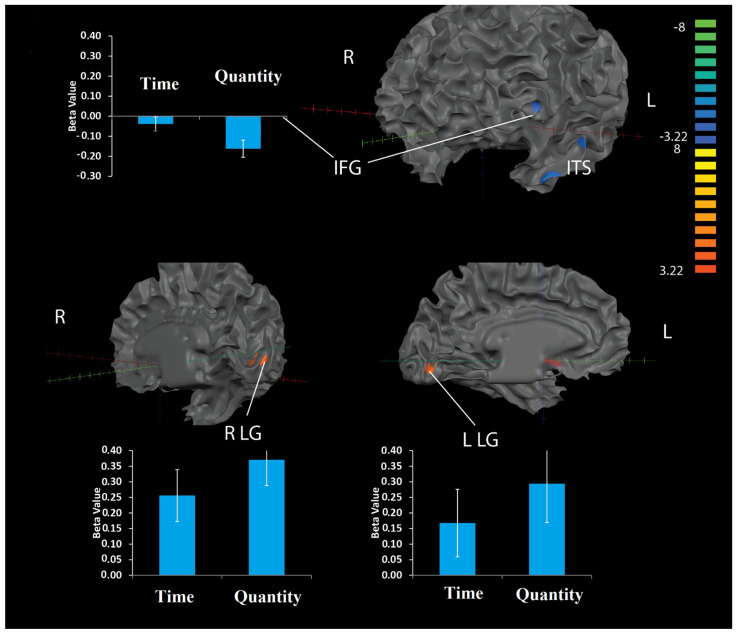
Brain activations related to CET numerical estimation > time. Illustrated are brain areas that showed differences between activity levels of numerical estimation compared to time estimation. Specifically, increased activation during numerical estimation was found in the left and right lingual gyrus (LG 14, −82, −2 and −6, −80, −6). Increased deactivation during numerical estimation was found in the right inferior frontal gyrus (IFG) (−43, −60, 19). Quantity = numerical estimation.

**Figure 6 brainsci-12-00104-f006:**
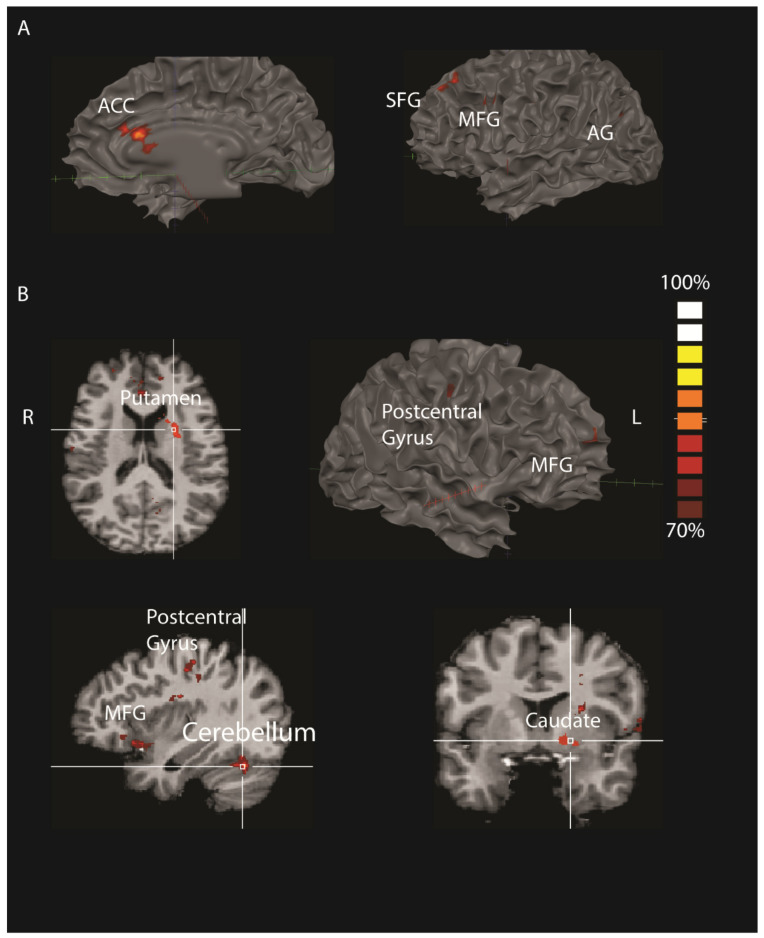
Multi-voxel pattern analysis of brain activations related to CET numerical quantity compared to time (**A**) and numerical quantity compared to weight (**B**). (**A**). Results of this analysis revealed differences in the quantity versus time categories in various frontal regions: left superior frontal gyrus (SFG), middle frontal gyrus (MFG), and right anterior cingulate cortex (ACC). Moreover, this analysis revealed differences in the left angular gyrus (AG). (**B**). Results of this analysis revealed differences in the quantity versus weight categories in the right and left middle frontal gyrus (MFG), the right postcentral gyrus, the left putamen, the left caudate, and in the right cerebellum.

**Figure 7 brainsci-12-00104-f007:**
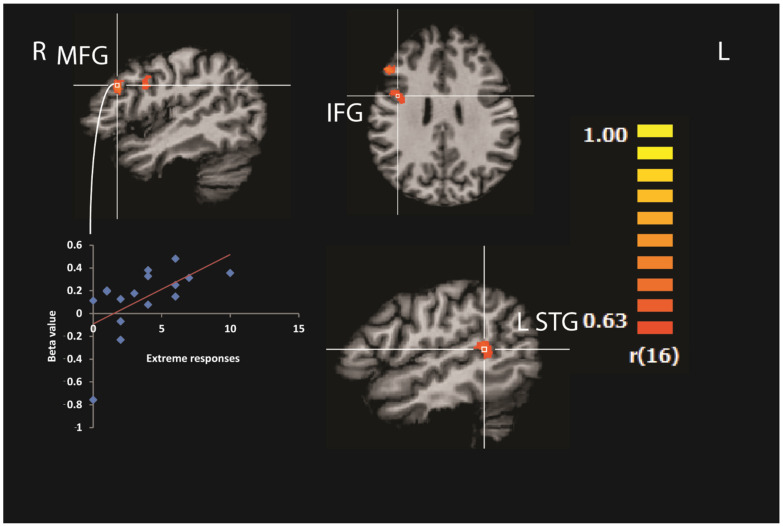
Correlation analysis between brain activation during cognitive estimation and extreme estimation responses. To reveal the brain regions that showed significant activation differences between the estimation categories, we calculated a whole-brain covariate model with the number of extreme CET responses. The results indicated positive correlations between the activity of the right inferior frontal gyrus (IFG), the right middle frontal gyrus (MFG), and the left superior temporal gyrus (STG) and extreme CET responses.

**Table 1 brainsci-12-00104-t001:** Brain Region by Contrast.

Brain Region	Brodmann Area	Coordinates			*t*	Cluster Size (Voxels)
		x	y	z		
CET > Exact Knowledge						
R supramarginal gyrus	40	59.02	−47.32	22.22	−4.2	1173
R caudate extending to anterior cingulate and precentral gyrus	6	9.33	11.54	27.47	7.8	39,576
R precentral gyrus	4	31.24	−20.82	53.45	5.7	6717
R lingual gyrus	18	0.84	−85.15	−0.86	5.4	5930
L R posterior cingulate	30	14.01	−52.1	6.64	5.2	1610
L cerebellum		−26.22	−53.53	−20.98	5	12,457
R cerebellum		32.32	−56	−28.23	5.7	6035
L lingual gyrus	18	−9.84	−52.7	5.47	4.9	2210
L insula extending to inferior frontal gyrus	13	−34.62	17.75	13.15	4.6	27,724
L angular gyrus	39	−54	−60.02	32.48	6.4	1065
Numerical estimation compared to weight						
No significant results						
Numerical estimation compared to time						
R lingual gyrus	17	13.7	−81.54	−1.58	5.8	2273
L lingual gyrus	18	−6.25	−79.58	−5.84	4.3	703
L inferior frontal gyrus	45	−43.14	−60.44	19.01	−4.6	877

Notes. *p* < 0.005 (cluster corrected for multiple comparisons, *p* = 0.05). Coordinates are in Talairach space. R = right, L = left, CET = cognitive estimation task.

**Table 2 brainsci-12-00104-t002:** Brain Region by Contrast, Multi-Voxel Pattern Analysis.

Brain Region	Brodmann Area	Coordinates			*t*	Cluster Size (Voxels)
		x	y	z		
Numerical estimation compared to weight						
R postcentral gyrus	43	57.44	−12.12	14.76	81	423
R cerebellum		34.33	−58.24	−18.95	94	1353
R cerebellum		35.39	16.59	−5.45	81	554
R medial frontal gyrus	9	1.6	37.27	26.41	94	1453
L putamen		−23.73	2.89	18.6	100	805
L caudate		−18.85	2.7	−0.98	88	684
Numerical estimation compared to time						
R inferior frontal gyrus	9	58.04	8.04	25.31	88	502
R middle frontal gyrus	8	45.6	26.61	41.04	88	403
R middle frontal gyrus	6	30.44	9.64	43.44	81	418
Thalamus		16.74	−26.53	−0.64	100	2179
R superior frontal gyrus	6	12.95	11.84	64.61	100	1095
R anterior cingulate	33	3.72	18.84	20.89	100	3282
R subcallosal gyrus	25	7.79	−12	−12.42	88	564
L superior frontal gyrus	8	−8.67	46.15	35.18	88	4011
L lingual gyrus	18	−15.9	−82.65	2.45	82	522
L angular gyrus	39	−42.09	−73.39	27.02	88	705
L anterior cingulate		−35.73	−2.04	28.19	94	494
L middle frontal gyrus	8	−48.4	13.72	41.36	100	1299
L precentral gyrus	4	−54.19	−15.9	37.31	88	644
L middle frontal gyrus	9	−57.56	20.24	26.58	88	439

Note. Cluster level > 400 voxels. Coordinates are in Talairach space. R = right, L = left.

**Table 3 brainsci-12-00104-t003:** Brain Regions Showing Significant Correlations with Extreme Estimations (Corrected using FDR).

Brain Region	Brodmann Area	Coordinates			*r*	Cluster Size (Voxels)
		x	y	z		
R middle frontal gyrus	46	55	27	29	0.78	546
R inferior frontal gyrus	9	46	6	27	0.74	757
L cingulate gyrus	32	−26	−12	35	0.84	1383
L superior temporal gyrus	22	−56	−48	11	0.81	1049

Note. Coordinates are in Talairach space. R = right, L = left.

## Data Availability

Data will be available upon request from the first author.

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
