# Peer review of "Understanding Estimations of Magnitudes: An fMRI Investigation"

_brainsci, 2022, doi:10.3390/brainsci12010104_

Round 1

Reviewer 1 Report

This fMRI study tackles an interesting question: what are the neural circuits involved in different magnitude estimations and in how far do they overlap or differ. The behavioural and fMRI data of 18 adults performing three types of estimation tasks (numerical, time and weight) are presented and compared with each other and with a control task (exact knowledge). Some interesting differences in brain activation patterns are found and discussed. In particular, the analyses related to extreme responses is interesting. Overall, the manuscript is adequately written but some points need to be clarified further.

Abstract

  • Please change the sentence stating that your results ‘demonstrate dissociations between discrete numerical estimations and estimations of continuous quantities’. Significantly higher or lower activation does not provide evidence for dissociations.

Introduction

  • I am struggling in parts with the logic of the introduction. For example, it does not necessarily follow from a consecutive acquisition of two skills over development that they should be dissociable in adults?
  • Numerous aspects of EF are mentioned (working memory, attention, cognitive control and planning) but then only the role of the DLPFC for EF is mentioned. Please include a proper short review of brain areas involved in the different aspects of EF.
  • The distinction between numerical estimation and continuous magnitudes needs to be motivated differently. Clearly, numbers are continuous too – so the concept of natural numbers and their use had to be made stronger. The examples don’t help because clearly you can talk about 1 hour, 2 hours, ect., so how is that different from 5 cookies?
  • The additional load of choosing a unit of measurement is an important (and potentially confounding !!) factor. I assume all three ranges offered as solutions for each problem used the same unit of measurement, correct?
  • State more clearly what the contribution of an understanding of the neural mechanism of CET would bring. Why is that important to know?
  • In line 139: it’s not clear to me what ‘explained resultant individual differences’ means here. Please clarify.
  • Using activation in prefrontal/orbital frontal cortices as a marker for EF and in inferior/posterior parietal cortex as a marker for numerical estimation is an overstatement. Surely this is an interpretation of results.
  • I am unsure what brain activity representations (line 144/145) are. Please clarify.

Material & Methods

  • Please describe the educational background of your participants (e.g., were they all university students, specific subjects?)
  • The baseline task is potentially problematic. The example encourages counting. This is not a good baseline.
  • While I appreciate the behavioral pilot to get percentiles for the CET, to call the results based on 48 students ‘norms’ is pushing it too far.
  • I don’t understand why participants were told ‘ any estimated answer would be correct’. Please clarify.
  • Please give more information about the event-related fMRI analyses. Which timewindow of the fMRI data was included? Did it include the question presentation (8s) and the presentation of the responses (3s)? In which case again the baseline task is problematic because of the overall faster RTs. Or was the event defined based on RTs in each trial? Please clarify which bits of the trials were included in the fMRI analyses.

Results

  • Please report the average percentiles (and SD) for each task
  • Please add clearer labels to Figure 4 and Figure 5
  • Why are the error bars so large for Quantity and Exact in LG in Figure 4?
  • Please correct for multiple comparison when reporting correlations

Discussion

  • The correlational analysis with extreme responses is very interesting and deserves more discussion

Typos/grammar

Line 72: stimuli -> stimulus

Author Response

Review 1

Abstract

  • Please change the sentence stating that your results ‘demonstrate dissociations between discrete numerical estimations and estimations of continuous quantities’. Significantly higher or lower activation does not provide evidence for dissociations.

Done, see line 21 in the abstract: "demonstrating different profiles of brain activations between discrete numerical estimations and estimations of continuous magnitudes.”

Introduction

  • I am struggling in parts with the logic of the introduction. For example, it does not necessarily follow from a consecutive acquisition of two skills over development that they should be dissociable in adults?

We have changed the introduction and refined our claims in the introduction.  

  • Numerous aspects of EF are mentioned (working memory, attention, cognitive control and planning) but then only the role of the DLPFC for EF is mentioned. Please include a proper short review of brain areas involved in the different aspects of EF.

Thank you for this comment, the current version of the manuscript includes a short review of brain areas involved in the different aspects of EF. Please see line 99 in the introduction.

  • The distinction between numerical estimation and continuous magnitudes needs to be motivated differently. Clearly, numbers are continuous too – so the concept of natural numbers and their use had to be made stronger. The examples don’t help because clearly you can talk about 1 hour, 2 hours, ect., so how is that different from 5 cookies?

We have changed the demonstration; see line 113 until 123

  • The additional load of choosing a unit of measurement is an important (and potentially confounding !!) factor. I assume all three ranges offered as solutions for each problem used the same unit of measurement, correct?

Yes, the unit of measurements, during the fMRI scan, was given to a participant. The participants did not freely chose the unit of measurements. Hence, the three ranges were given with the same unit of measurements. Here is the example that we use in the text: (e.g., for a weight question – 1. Below 112 kg; 2. Between 112-360 kg; 3. Between 361-520 kg; 4. More than 520 kg). As one can see, the Kg is the same in all the possible responses.   

  • State more clearly what the contribution of an understanding of the neural mechanism of CET would bring. Why is that important to know?

We added this to the current study see line 148.

  • In line 139: it’s not clear to me what ‘explained resultant individual differences’ means here. Please clarify.

Done. We have made it clearer.

  • Using activation in prefrontal/orbital frontal cortices as a marker for EF and in inferior/posterior parietal cortex as a marker for numerical estimation is an overstatement. Surely this is an interpretation of results.

We have refined these sentences. Please see line 155. 

  • I am unsure what brain activity representations (line 144/145) are. Please clarify.

We have changed it to cognitive representation.

Material & Methods

  • Please describe the educational background of your participants (e.g., were they all university students, specific subjects?)

Done – the information was added on line 172.

  • The baseline task is potentially problematic. The example encourages counting. This is not a good baseline.

We have change the example.

  • While I appreciate the behavioral pilot to get percentiles for the CET, to call the results based on 48 students ‘norms’ is pushing it too far.

We have change the terminology and deleted the term norms.

  • I don’t understand why participants were told ‘ any estimated answer would be correct’. Please clarify.

We have added more details about it; please see line 226- 227.

  • Please give more information about the event-related fMRI analyses. Which time-window of the fMRI data was included? Did it include the question presentation (8s) and the presentation of the responses (3s)? In which case again the baseline task is problematic because of the overall faster RTs. Or was the event defined based on RTs in each trial? Please clarify which bits of the trials were included in the fMRI analyses.

We added the information to the text; please see lines 235 to 237.

Results

  • Please report the average percentiles (and SD) for each task

Done please see Table S1.

  • Please add clearer labels to Figure 4 and Figure 5

Done.

  • Why are the error bars so large for Quantity and Exact in LG in Figure 4?

It was a mistake - we have corrected it.

  • Please correct for multiple comparison when reporting correlations

The correlations were corrected for multiple comparisons in similar way to the whole brain analysis. We have added it to the text.

Discussion

  • The correlational analysis with extreme responses is very interesting and deserves more discussion

We added more discussion to the text -  please see line 441until 444.

Reviewer 2 Report

In this study, Ashkenazi and colleagues performed an fMRI study of human participants and examined activation of the brain during quantitative estimation. They identified long-range brain networks that process discrete numerical estimation differently from some types of continuous estimation, and reach the conclusion that separate pathways process different types of continuous magnitude estimation, partly related to the units of measurement. I think it is an interesting study, and I only have the following minor comments to make:

Minor:

  1. Abstract, line 14: “latter”, rather than “later
  2. Introduction, lines 60 and 72: “stimulus”, rather than “stimuli”
  3. Introduction, line 83: Change to “CET was originally designed…”
  4. Introduction, line 89: Change to “important”, or “of importance”.
  5. Lines 231 & 355: Remove the word “by” from “by using”.
  6. Lines 412-414: The sentence “However…exact knowledge” is confusing and should be re-structured. It seems to be treating the IPS as part of the ventral visual stream, while line 409 treats it as part of the dorsal visual stream.
  7. Line 442: “attended focus of attention” should probably change to “current focus of attention”.

Author Response

Reviewer 2

In this study, Ashkenazi and colleagues performed an fMRI study of human participants and examined activation of the brain during quantitative estimation. They identified long-range brain networks that process discrete numerical estimation differently from some types of continuous estimation, and reach the conclusion that separate pathways process different types of continuous magnitude estimation, partly related to the units of measurement. I think it is an interesting study, and I only have the following minor comments to make:

Minor:

  1. Abstract, line 14: “latter”, rather than “later

Done.

  1. Introduction, lines 60 and 72: “stimulus”, rather than “stimuli”

Done.

  1. Introduction, line 83: Change to “CET was originally designed…”

Done.

  1. Introduction, line 89: Change to “important”, or “of importance”.

Done.

  1. Lines 231 & 355: Remove the word “by” from “by using”.

Done.

  1. Lines 412-414: The sentence “However…exact knowledge” is confusing and should be re-structured. It seems to be treating the IPS as part of the ventral visual stream, while line 409 treats it as part of the dorsal visual stream.

Thank you; we changed the ventral visual stream to the parietal lobe.

  1. Line 442: “attended focus of attention” should probably change to “current focus of attention”.

Done.